# The Digital Authoritarian: On the Evolution and Spread of Toxic Leadership

**Brian L. Ott** *  **and Carrisa S. Hoelscher**

Department of Communication, Media, Journalism and Film, Missouri State University, Springfield, MO 65897, USA; choelscher@missouristate.edu
* Correspondence: brianott@missouristate.edu

**Abstract:** Employing a critical approach typical of humanities-based research, this article investigates the changing nature of toxic leadership in our digital world. Drawing on the perspective of media ecology, which asserts that the prevailing communication technologies at a given moment create the social conditions that, in turn, condition us, the authors illustrate how the digital logics of publicity, intransigence, impertinence, and impulsivity remake the contours of leadership. Based on a critical case study of Elon Musk's public management of Twitter, which has subsequently been rebranded as "X", it is argued that the four digital logics transform toxic leadership into digital authoritarianism, an unabashed form of authoritarian rule. A concluding section of the essay explores the implications of this evolution for traditional categories of leadership; the importance of attending to communication technologies in leadership research; and the individual, institutional, and social harms of digital authoritarianism.

**Keywords:** digital society; public management; toxic leadership; authoritarianism; digital logics; media ecology; Elon Musk; Twitter; X

---



## 1. Introduction

Ours is a digital world. As such, we are almost entirely dependent on digital technologies for news, entertainment, communication, and commerce. On one hand, life in a digital world offers unparalleled convenience, connectivity, and access to information, potentially empowering us in unprecedented ways. On the other hand, it means that screens increasingly mediate social interaction, algorithms influence the nature of our experiences, and digital environments blur the boundary between the material world and a virtual one. In short, digital technologies pervade and shape nearly every aspect of our social lives.

Thus, it comes as no surprise that digital technologies are transforming the nature of work. They are, for instance, encouraging entrepreneurialism, eliminating or streamlining repetitive tasks, enhancing productivity and efficiency, and enabling remote work, flexible schedules, and cross-cultural and interorganizational collaborations. But, perhaps less obviously, digital technologies are also giving rise to new models of leadership. For one thing, digital technologies make leadership—like virtually everything today—decidedly more public.

Increasingly, leaders at every level—from heads of state to corporate CEOs to the line manager at a local Starbucks—may choose to conduct business and "lead" in full view of the public, largely on social media platforms. During Donald Trump's presidency, for instance, one rarely had to wonder what the president was thinking about or what his approach to a particular issue might be, as he consistently broadcast both on Twitter [1]. Similarly, one need not imagine what it might be like to work for Elon Musk, as his management of Twitter, which he recently rebranded as "X", has unfolded one tweet at a time right before our eyes.

Donald Trump and Elon Musk, one a politician and the other the world's wealthiest person, are not random examples. As Robert Reich observes, "both represent the emergence

of a particular . . . personality in the early decades of the 21st century: the wildly disruptive narcissist" [2] (para. 10). But they share, we argue, more than a particular personality; they share an evolving style of management that is spreading rapidly in politics and business, as well as in educational and religious contexts. Our central goal in this study is twofold: to chart the contours of that style, which we have dubbed digital authoritarianism, and to illustrate how it operates through a critical case study.

In terms of its contours, we specifically argue that digital authoritarianism is a unique combination of: (1) the destructive behaviors and dysfunctional psychological traits of toxic leaders; (2) the ideology of authoritarianism; and (3) the underlying structural logics of digital technologies. In terms of our supporting case study, we have selected Elon's Musk takeover and leadership of Twitter. We made this decision because Elon Musk offers a particularly clear example of public corporate leadership, regularly making and communicating management decisions on his recently acquired social media platform. We refer to that platform throughout the essay as Twitter since we had mostly completed our study by the time Musk rebranded it as X.

To facilitate these goals, our essay unfolds in four stages. First, we review and reflect on the relevant literature regarding toxic leadership, authoritarianism, and media ecology. Second, we propose a critical approach uniquely suited for investigating digital authoritarianism. Third, we analyze Elon Musk's leadership both of and on Twitter as an example of digital authoritarianism, highlighting the ways it remakes toxic leadership. Fourth, we discuss the broader implications of the spread of this management style and consider its damaging personal, organizational, and social consequences.

Before proceeding, we wish to acknowledge that not everyone regards authoritarianism as damaging or dangerous. Both Donald Trump and Elon Musk have amassed vast followings of individuals who fervently believe that their style of management has led to predominantly positive outcomes. Here, we invoke the observation first made by Theodor Adorno et al. [3], who argued that a certain personality type exists that will find authoritarianism not only acceptable, but preferable for addressing societal problems. Bill Jones cautions that "there are foreboding signs of Adorno's warnings coming to pass in the US", as it increasingly abandons democratic norms [4] (p. 34). Again, we recognize not everyone views this "authoritarian slide" as problematic or novel, and, in fact, one of the reviewers of this essay suggested that democracy itself may be a historical "aberration". Our own view is that—aberrant or not—democratic values are worth defending, and that to do so we must have a thorough understanding of how the features of our digital world amplify authoritarian strategies.

## 2. Literature Review and Reflection

Toxic leadership is new neither in practice nor in theory [5]. But it is, we contend, evolving in practice, which invites a rethinking of the theory. That evolution, we further argue, is a product largely of changes in our communication environment and the technologies that animate it. Before looking at how toxic leadership is changing in a digital world, we briefly review the history of the concept.

As Steven Walker and Daryl Watkins recount, the political scientist Marcia Wicker introduced the phrase toxic leader in 1996 [6], though Jean Lipman-Blumen renewed interest in the topic and heightened its popularity with her 2004 book, The Allure of Toxic Leaders. In her book and elsewhere, Lipman-Blumen defines toxic leadership as, "a process in which leaders, by dint of their destructive behavior and/or dysfunctional personal[ity] characteristics inflict serious and enduring harm on their followers, their organizations, and non-followers, alike" [3] (p. 36). Expanding on this definition, Asha Bhandarker and Snigdha Rai observed, a "leader can be considered toxic if [their followers are] physically or psychologically harmed by the leader's actions and it creates long-lasting impairment in the subordinates" [7] (p. 66).

As these definitions suggest, the scholarship on toxic leadership has tended to view it through the lens of its effects or consequences on followers. Similarly, the research

has sought to understand why toxic leadership is compelling to many followers. As Jean Lipman-Blumen has noted, "followers not only tolerate, but ... often prefer, and sometimes even create toxic leaders—in for-profit corporations, non-profits, government, even educational and religious institutions" [8] (p. 1). People are attracted to toxic leaders, according to Lipman-Blumen, for six primary reasons; they (1) appeal to deep psychological needs, (2) ease existential anxiety, (3) provide order in a chaotic world, and foster a sense of (4) belonging, (5) belief, and (6) purpose. Charlice Hurst et al. proposed a seventh reason some employees follow toxic leaders, which is that they themselves show signs of psychopathy, and employees with high primary psychopathy are more likely to flourish than their peers under toxic leaders [9].

While much research focuses on the motivations of followers, scholars have identified a series of destructive behaviors and dysfunctional personality traits typical of toxic leaders [10,11]. Some of the key destructive behaviors in which toxic leaders engage include misleading, lying, undermining, stifling, silencing, demeaning, demoralizing, bullying, intimidating, coercing, marginalizing, scapegoating, disenfranchising, and favoring. In addition to these behaviors, toxic leaders also exhibit a series of related dysfunctional behavioral traits, including insatiable ambition, narcissism, self-aggrandizement, arrogance, and a lack of honesty, integrity, trustworthiness, transparency, empathy, and self-reflection.

Summarizing these behaviors and personality traits, Birol Başkan writes, "There are three critical elements of this destructive style of leadership: an apparent lack of concern for the well-being of subordinates, a personality or interpersonal technique that negatively affects the organizational climate, and a belief by subordinates that the superior is motivated primarily by self-interest" [12] (p. 98). Inasmuch as all these elements run counter to prevailing understandings of effective leadership, toxic leadership can be understood as incompetent leadership combined with abuse. Put another way, toxic leadership is the exercise of abusive incompetence in a leadership position, and thus, at least in the extreme, not really leadership at all.

Due to the deficit of leadership behaviors associated with toxic leadership, toxic leaders lean heavily on authority to achieve their aims, which partially explains why toxic leaders are prone to authoritarianism. While some scholars regard authoritarianism as its own style of leadership [13,14], we posit that it is better understood as a mode of ruling rather than leading. Authoritarians, by definition, exert absolute authority and control and demand unconditional obedience [15]. Leaders (even toxic ones), by contrast, do not exercise absolute authority. Indeed, toxic leaders typically do not embody all the destructive behaviors and dysfunctional personality traits of toxic leadership or enact them in all contexts [8], though the more of these traits a leader possesses and the more routinely they are enacted, the more authoritarian that leader is likely to be.

In the political sphere, the spread of authoritarianism around the globe has accelerated in recent years, capturing the attention of commentators and scholars alike. In one worldwide assessment, Sarah Repucci and Amy Slipowitz note that the number of countries with declines in democratic indicators has far outpaced the number of countries with democratic improvements over the past 16 years [16]. The systematic violation of democratic norms in favor of authoritarian rule has dire consequences for citizens, societies, and international relations, and understanding these consequences will become increasingly important if current trends continue [17].

While authoritarianism shares commonalities with toxic leadership, as well as other recognized styles of destructive leadership [18], its central concern with rulership rather than leadership makes it distinctive. Because authoritarianism is premised on centralized power, authoritarian rulers often have limited or no accountability. In short, they are neither responsible for their decisions nor accountable for their actions. They also exercise far greater control over the flow of information, often eliminating any possibility for discussion, let alone dissent. Indeed, one of the key differences between toxic leaders and authoritarian rulers is that authoritarians do not, properly speaking, have followers; they have subjects, and those subjects have limited rights. Subjects may, of course, openly (even earnestly)

support an authoritarian ruler, even a brutal one, though this is typically done out of fear or favor and is tied to a cult of personality.

Thus, while authoritarianism and toxic leadership are distinctive, we maintain that toxic leadership is evolving in the direction of authoritarianism. To understand why and how this is happening, we need to turn to the subject of media ecology. The central premise of this essay, as well as the field of media ecology generally, is that communication technologies are not merely tools in our social environment, but constitute the very social environment we inhabit, thereby conditioning our habits of mind [19]. Alternatively stated, media ecologists posit that the prevailing technologies of communication—the media forms that dominate at a given historical moment—shape and influence not so much what we know (i.e., our attitudes and beliefs about the world), but how we know (i.e., the way we process information and make sense of the world).

With respect to the topic of leadership, we are interested in two questions. First, how has the development and widespread use of digital technologies altered thinking (again, not what leaders think, but how leaders think)? In answering this question, we suggest that—among other habits of mind—some workplace leaders increasingly view employees as subjects to be ruled rather than followers to be led. Second, how have digital modes of consciousness or habits of mind altered the character and conduct of leadership? To answer the first of these questions, we review the relevant literature on the structural logics of digital media. Answering the second question is the central aim of our analysis.

According to media ecologists [20], all media forms have relatively fixed and distinct structural traits. In other words, they are designed and engineered to operate in particular ways. These structural traits or biases mediate human experience through repeated use, effectively training us to process the world in certain ways. Scholars generally divide human history into three major periods [21,22]: primary orality, literacy (further subdivided into writing and print cultures), and the electronic era (further subdivided into analog and digital cultures). Based on the structural biases of the prevailing media forms, each one of these eras gave rise to different modes of thought and expression.

For instance, we know that the experience of our social environment and, subsequently, thought in the period of primary orality, which existed before the invention of writing, was aggregative, immediate, participatory, and situational, while during the print era it was more analytical, interior, isolated, and distanced [22]. Today, in a digital world, experience and thought tend to be more affective, connected, interactive, and simulated. Brian Ott argues that digital media has three defining structural traits: digitality, algorithmic execution, and efficiency. These traits, in turn, foster the logics of intransigence, impertinence, and impulsivity [23] (p. 8).

Intransigence, which arises from digital media's basis in binary code, trains us to see the world in simple, dichotomous, and dogmatic ways. Impertinence, which arises from the programmed nature of computers, conditions us to be insensitive and unresponsive to others, while impulsivity, which is related to the efficiency of microprocessors, invites us to act affectively rather than analytically. To these, we would add a fourth logic, publicity, as digital media ensure that we are chronically online. We offer a more detailed discussion of these traits in our analysis. In sum, Table 1 highlights the key traits of toxic leadership, authoritarianism, and digital technology characteristic of digital authoritarianism.

**Table 1.** Constituent elements of digital authoritarianism.

| | Toxic Leadership | Authoritarianism | Digital Technology |
|---|---|---|---|
| **Central Concern** | Abusive Incompetence | Rulership | Structural Logics |
| **Key Traits** | • destructive behaviors • dysfunctional personality traits | • centralized power • controlled flow of information • no accountability • limited rights • silenced dissent • cult of personality | • intransigence • impertinence • impulsivity • publicity |

## 3. Critical Approach

Because we are operating within a critical (rather than empirical) research paradigm, we adopt a broad approach or "view" rather than a rigid "method". In this section, we outline our unique approach to studying digital authoritarianism. But before doing so, we need to review briefly why a novel approach is warranted. Traditionally, research on leadership in organizational communication and management has studied it independent of the technologies of communication that create the very social environment we inhabit. This oversight blinds us to the habits of mind specific to the social environment at a given historical juncture. In other words, while scholars have explored how technologies are used by leaders, they have generally ignored how oral, chirographic, typographic, analog, and digital cultures all foster different models or styles of leadership.

But a digital culture is fundamentally different from an oral or print culture, and it produces different ways of thinking both by and about leaders. Take toxic leadership, which existed long before digital technologies dominated all aspects of our social lives. In a pre-digital world, an employee with a toxic boss had to endure their boss's destructive behaviors and dysfunctional personality traits at work, but those behaviors and traits were not on full display for the public. Nor were online trolls or sycophants—a few of whom perhaps work at the same organization—liking, praising, and encouraging those toxic behaviors publicly.

A digital culture has changed this. A toxic boss today, who perhaps is particularly given to the dysfunctional personality trait of narcissism, enthusiastically posts on social media about the restructuring of their organization. Users (perhaps dozens, hundreds, or even thousands)—most of whom do not work at this organization and who have no way of knowing this decision was made in an entirely top-down manner that silenced and disenfranchised employees—praise the realignment. In this instance, the symbolic violence visited upon employees by the manner of the decision is revisited upon the employees by the social media post and subsequent responses. This phenomenon is evident, for instance, in the reorganization that took place at Twitter following Elon Musk's takeover.

Moreover, the toxic leader, seeing and feeling affirmed by the public praise for their destructive behavior, is encouraged to engage in similar behavior again. But perhaps the most significant point about the preceding example is that it focuses mainly on the logic of publicity and the way digital technologies have made leadership more public in the digital world. As we highlighted previously in the literature review, the logics of intransigence, impertinence, and impulsivity, in addition to the logic of publicity, are all remaking the character of toxic leadership in the digital era. It is crucial, therefore, that scholars who study leadership attend carefully to the structural features of digital technologies and highlight the styles of leadership they foster.

In response to these issues, our approach to studying digital authoritarianism is twofold: first, we examine how the structural logics of digital communication technologies influence the thinking (i.e., the habits of mind) and, consequently, the behaviors of toxic leaders; and second, we demonstrate the subsequent style of leadership through a critical case study, namely Elon Musk's takeover and leadership of Twitter. Musk is hardly the only example of this evolving style of leadership. But he is, we think, a particularly high profile and clear example of how digital media remakes toxic leadership. Given the extraordinarily public character of his leadership, we conducted our analysis based on publicly available data, attending to (1) public accounts of Elon Musk's leadership as reported in news media and (2) examining several of Musk's actual tweets as evidence of leadership practices and behavior. These data are attached as supplementary materials under the title, "Elon Musk Leadership Action Chart".

We recognize that some readers may object to our approach on the grounds that it conflates an analysis of media coverage of Elon's Musk's leadership with an analysis of Musk's actual leadership. But from our perspective, such a critique fails to understand how digital media technologies have obliterated that distinction. Digital authoritarianism as a style of leadership is significant and warrants our attention precisely because it is mediated

by digital technologies. Thus, any serious attempt to give an account of it would, out of necessity, require attention to its fundamentally mediated character. To put it another way, Elon Musk's "public" management of Twitter is his management of Twitter. One of our central arguments is that there is no longer a meaningful distinction between the two.

## 4. Critical Analysis

The core argument of this essay is that our digital world and, more precisely, the communication technologies that have given rise to that world are producing and proliferating a new style of leadership known as digital authoritarianism. This evolving style of leadership encompasses the destructive behaviors and dysfunctional personality traits typical of toxic leadership. But it subsumes those behaviors and traits to the prevailing logics of digital media, which collectively function to make toxic leadership more authoritarian. Basically, the inescapability of digital media and its structural biases are transforming toxic leadership into digital authoritarianism by enlisting its central features in service of publicity, intransigence, impertinence, and impulsivity.

Given this transformation, we have two primary goals in this portion of the essay. Our first goal is to clarify the structural biases and attendant logics of digital media, paying particular attention to the ways those logics incorporate and remake the destructive behaviors and dysfunctional traits of toxic leadership. There is not, however, a simple one-to-one relationship between the four digital logics and various features of toxic leadership. On the contrary, each of the digital logics enlists a constellation of toxic behaviors and traits, and several of the logics enlist the same behaviors and traits. For instance, the toxic leadership trait of arrogance serves all four digital logics. Our second goal is to illustrate the operation of these logics in Elon Musk's leadership of Twitter by drawing on news media accounts of his management practices and analysis of his tweets.

### 4.1. Publicity: Self-Promotion and Self-Interest in an Attention Economy

If our digital world has a master logic, a logic through which all other logics pass, it is publicity. The logic of publicity combines self-promotion and self-interest with a near continuous desire to remain in the public eye. Like all digital logics, publicity is a consequence of the inherent structural biases of digital communication technologies, though it receives its fullest expression on social media platforms like Facebook, Instagram, YouTube, LinkedIn, TikTok, and Twitter.

The development and widespread adoption of digital communication technologies have eroded the traditional distinctions between our personal, professional, and public lives. Increasingly, thoughts and actions exercised in private and professional settings (in front of a limited audience) are shared publicly, sometimes by choice and sometimes not. Either way, the increasingly public nature of our personal and professional lives is a consequence of two key structuring principles associated with digital media: easy access to the modes of production and digital tracking.

First, unlike legacy media, whose ownership and control were concentrated in the hands of a select few, the ubiquity of digital media, especially mobile telephony, make anyone a potential purveyor of mass information. By decentering the modes of production, digital media turn users into producers, democratize the dissemination of information, and invite "sharing." Second, digital media, which "track" every keystroke, website, and user preference, condition us to unreflexively accept the reality of continuous surveillance, making us even less likely to police traditional boundaries between our personal, professional, and public lives.

Simply put, the desire to be seen is deeply embedded in the structural logic of digital media and social media in particular. On these platforms, people curate and share carefully crafted versions of themselves and their lives, often highlighting achievements and successes. The culture of following, liking, and commenting on social media platforms, which is engineered to produce regular hits of dopamine, further incentivizes people to behave in self-promotional and self-interested ways. This raises the question of what happens

when the digital logic of publicity infects a leader's habits of mind and, consequently, management practices?

Because publicity reflects the master logic of the digital age, it tends to be expressed more clearly in the general management philosophy of a leader than in specific leadership actions, though there are instances when it is acutely evident in those as well. As such, it is not so much what Elon Musk tweets (e.g., the content of tweets) that exemplifies the logic of publicity but how and how much he tweets (e.g., the form of tweets). As Linda Chong et al. wrote in The Washington Post, "Elon Musk is a Twitter superuser. He has tweeted more than 19,000 times since joining the platform 13 years ago. This year, he has tweeted an average of six times a day. At least 150 of those posts are about Twitter itself" [24] (para. 1). Similarly, Kari Paul, writing in *The Guardian*, described Musk as a "prolific user" [25].

While the specific content of Elon Musk's tweets since taking over as CEO regularly introduced policy changes at Twitter, the need to announce them publicly on the platform appears to reflect a deeper psychological impulse. As Jesse Lehrich, co-founder of the advocacy group Accountable Tech, told Billy Perrigo at Time in April 2023, "Throughout his career, Musk has had an almost pathological need to promise grand visions and make himself the center of attention. He's very Trumpian in his need to capture media attention with constantly-shifting promises, which everyone in the media covers" [26] (para. 2).

Elon Musk not only routinely promises grand visions such as rebranding Twitter as "X", telling the world it will become the "everything app", but he also often makes those pronouncements with little warning or context, which heightens their surprise or shock value and ensures wider circulation. After all, the algorithms that govern social media platforms prioritize content that garners more attention, which encourages individuals to act in ways that attract a larger audience, and what better way to gain attention than to violate norms and expectations.

Tomas Chamorro-Premuzic refers to this practice as "grandiose exhibitionism" in his 2023 book, I, Human, and suggests it is, "One of the key facets of narcissism . . . which is characterized by self-absorption, vanity, and self-promotional impulses and is especially well-suited to a world in which human relations have been transferred almost entirely to digital environments" [27] (p. 85). Narcissism is, of course, one of the primary dysfunctional personality traits of toxic leaders, and "More than anyone else, narcissistic individuals feel the constant need to be the center of attention, even if the means to achieving this is to engage in inappropriate, awkward, or eccentric interpersonal behaviors" [27].

The impulse to be the center of attention animates virtually everything Elon Musk does. Indeed, one could say that publicity is Musk's leadership philosophy. Comparing Musk to Donald Trump, Robert Reich offered the following assessment in The Guardian, "Both are indefatigable self-promoters. Both are billionaires, but they are not motivated primarily by money. Nor are they fueled by any larger purpose, principle or ideology. Their singular goal is to imprint their giant egos on everyone else" [2] (paras. 12–14). In fact, many commentators believe that ego and self-interest are what drove Musk to purchase Twitter in the first place. "Elon Musk", Tomas Chamorro-Premuzic writes, "wasn't content with monopolizing so much attention on Twitter, so he offered $44 billion to buy the entire business (then getting even more attention by pulling out [or at least trying to] of the agreed deal" [27] (p. 85).

Adopting the logic of publicity as a guiding leadership philosophy naturally influences individual decisions, of course. One of the clearest examples of this with regard to Elon Musk's management of Twitter was his response "when his Super Bowl tweet performed worse than . . . president [Biden]'s" [28] (para. 1). Writing for Vox, Shirin Ghaffary reported, "Musk, apparently livid because his tweets about the Super Bowl were getting fewer views than President Joe Biden's, flew to Twitter's headquarters and ordered engineers to change the algorithm underlying Twitters main product to boost his own tweets above everyone else's" [29] (para. 4).

While Elon Musk's embodiment of the digital logic of publicity may seem victimless, his alteration of Twitter's boosting algorithm as a result of that logic suggests otherwise.

It highlights that Musk is motivated not by what is in the best interest of his employees and even less so by what is in the best interest of Twitter, but in his own self-interest. Leaders who make decisions based on self-interest make bad business decisions that harm both employees, often lowering morale, and the organization they serve, often hurting the bottom line. Musk's decision to purchase Twitter to feed his own ego caused him to grossly overpay for the platform and his self-interested business decisions after becoming CEO chased away many advertisers on Twitter and undermined stakeholder confidence.

*4.2. Impertinence: Dichotomous and Dogmatic Thinking*

The second logic associated with digital media is intransigence, which describes a habit of mind involving simple, short-sighted, either-or, inflexible thinking. Basically, impertinent thinkers lack cognitive complexity, which in the case of leaders prompts them to arrogantly propose reductionistic solutions to complex problems and to eschew responsibility for consequences by blaming and scapegoating others when things go awry. This logic arises out of digital media's most basic structural feature: binary code. In contrast to humans, who have traditionally processed their environment and everything in it as a continuous and interconnected stream of sensations, images, and words [30], "A computer thinks—if thinking is the right word for it—in tiny [discrete] pulses of electricity, representing either an 'on' or an 'off' state, a zero or a one" [31] (p. 14).

Repeated exposure to binary code, according to Brian Ott, "unconsciously urges us to divide our otherwise contiguous world into discrete units, units that are not only separate but also fundamentally opposed. In binary code, one and zero are opposing states; when the human mind tries to make sense of our analog world in this manner, it contributes to dichotomous, absolutist, and inflexible thinking" [23] (p. 8). Just as the logic of publicity conditions us to want to be seen, the logic of intransigence conditions us to see in black-and-white. The prevalence of intransigent thought in digital environments is evidenced by the speed and ease with which issues—both serious and seemingly innocuous—become irrevocably polarized.

The logic of intransigence is also evident in a wide range of Elon Musk's management decisions at Twitter, including his advocacy of free speech absolutism, his subsequent devaluing of content moderation on the platform, his reinstatement of Donald Trump's Twitter account based on an online poll, his removal of the legacy blue verified checkmark system [32], and his hostility toward research on the platform [33]. Indeed, the short-sightedness of these and other decisions led to a wide array of problems at Twitter after Elon Musk took over, not the least of which was advertisers abandoning the platform.

As Clare Duffy reported for CNN in February 2023, "More than half of Twitter's top 1000 advertisers in September were no longer spending on the platform in the first weeks of January . . . After Musk completed his takeover of the company in late October, advertisers began to worry about the safety and stability of the platform given his plans to cut staff and relax content moderation policies" [34] (para. 1). The decline in advertising revenue, which "Musk has admitted . . . is down by 50%" [35] (para. 7), is significant because historically advertising "has made up 90% of Twitter's revenue" [36] (para. 25).

The clearest example of intransigent thinking by Elon Musk involves his advocacy of free speech absolutism and subsequent relaxing of content moderation. In a 5 March 2022, tweet, Musk proclaimed himself "a free speech absolutist", a statement the news media has circulated widely without objection from Musk ever since. In an interview for Time, Jason Goldman, a member of Twitter's corporate board from 2007 to 2010, described Musk's free speech absolutism as "naïve" for failing to understand the political and legal complexities of the issue [37] (para. 15). But Musk did more than articulate this simple-minded position and dogmatically refuse to moderate it, he acted upon it.

Specifically, "Musk", wrote Billy Perrigo, "fired many members of Twitter's platform safety team just days before the U.S. midterm elections, . . . removed bans on dozens of accounts including Neo-Nazis, and disbanded the platform's already-existing Trust and Safety Council" [26] (para. 3). Even before implementing these changes, Elon Musk's

declaration of being a free speech absolutist had prompted a proliferation of hate speech on the platform. As The Guardian reported in October 2022, "many began testing the limits of the site just hours after the billionaire took the helm. . . . dozens of extremist profiles—some newly created—circulated racial slurs and Nazi imagery while expressing gratitude to Musk. And researchers found a surge in new followers flocking to the accounts of high-profile rightwing figures in the 24 h after Musk took over" [38] (para. 14).

To combat the loss of advertising revenue created by his absolutist position on free speech, weakening of moderation standards, and subsequent proliferation of hate speech, Elon Musk decided to replace the legacy blue checkmark system, which functioned to verify users were who they claimed to be, with a paid checkmark system. But Musk, who had fired nearly half of the staff at Twitter shortly after being hired, had no plan to verify users who paid for the checkmark, which led to a host of difficulties. In his reporting for CNN, Brian Fung captured the chaos that ensued:

> Twitter users awoke Friday morning to even more chaos on the platform than they had become accustomed to in recent months under CEO Elon Musk after a wide-ranging rollback of blue check marks from celebrities, journalists and government agencies. The end of traditional verification marked the beginning of a radically different information regime on Twitter, one highlighted by almost immediate impersonations of government accounts; the removal of labels previously used to identify Chinese and Russian propaganda; and a scramble by the company to individually re-verify certain high-profile figures such as Pope Francis. [39] (paras. 1–2)

Elon Musk's management of Twitter consistently suggests that he sees the world in very simplistic, black-and-white terms. As such, he proposes preposterously reductionistic solutions to complex problems like content management. But he also lacks the self-awareness and reflexivity to take responsibility for his colossal missteps, choosing like many toxic leaders to blame others. In an interview for CNBC Make It, Harvard leadership author and expert Bill George told the outlet, "If you had to write a case study on an example of a really poor takeover of an organization, Elon Musk's takeover of Twitter would fit that perfectly well. . . . I don't think he understands social media" [40] (para. 2).

*4.3. Impertinence: Insensitive and Inhumane Behavior*

The third logic of digital media is impertinence; it reflects a habit of mind that favors callousness over compassion and cruelty over caring. The logic of impertinence derives from the essential nature and function of digital media. Digital media technologies are unaware, unreflexive, and dispassionate machines (even if they are sometimes programmed to simulate or feign otherwise). Their operating logic—their mode of information processing—is cold (unfeeling) and calculated (algorithmically programmed and mathematically executed). As such, according to Henry Perkinson, digital communication technologies are incapable of ethics or morality [41]. These machines are not immoral; they are literally *a-moral*, meaning they are entirely without morality.

In contrast to digital media, human beings are self-aware, reflexive, and qualitative decision-makers, equipping them with the unique capacity for ethics and morality. "So", queries Brian Ott, "what happens when the human mind tries to 'process' [and act on] information like a computer? If digitization and the logic of binary code foster an intransigent mind, one that is both dichotomous and dogmatic, algorithmic execution within a closed system habituates an impertinent mind, one that is increasingly insensitive and unresponsive" to the thoughts, feelings, and needs of others [23] (p. 9). In short, digital media invite persons to become more machine-like.

In the case of leaders, the logic of impertinence urges decision making that shows little concern or regard for what others think and feel. Much like authoritarianism, it rarely involves the solicitation of input (for anything other than show) and even more rarely alters course in response to feedback. It is rooted in an attitude of "I know best" and, consequently, does not hold itself as answerable or accountable to anyone. In keeping

with this uncaring and unresponsive attitude, digital authoritarians enact a wide range of toxic behaviors in positions of leadership, including but not limited to stifling, silencing, bullying, intimidating, demeaning, coercing, marginalizing, and disenfranchising. These behaviors, along with others that demonstrate a lack of human empathy, manifest widely in Elon Musk's management of Twitter.

The logic of impertinence was evident from the start of Musk's leadership both in the act of mass layoffs and in the manner in which those layoffs were carried out. As Johana Bhuiyan reported for The Guardian, "In the second week, nearly half of the company's workforce were laid off with little notice, prompting some to . . . file a class-action lawsuit alleging Elon Musk violated California labor law" [42] (para. 6). Those who remained were, according to Shana Lebowitz at Business Insider, not treated much better: "Shortly after Musk took the helm, some employees received instructions to work 12-hour shifts, seven days a week, without being told whether they would receive overtime pay or time off, CNBC reported. At the same time, Musk started ranking employees against one another" [43] (para. 7).

Writing for Forbes, Bryan Robinson offered this assessment: "Experts on workplace leadership assert that so far Musk's leadership style is headed in the wrong direction. . . . Musk is treating people like collateral damage instead of human beings, forgetting basic human decency in the way he's handling the layoffs" [44] (para. 2). Just a few weeks later, Eli Sopow rendered a similar judgment at The Conversation, noting that, "Musk's cold, impersonal approach to management and leadership is antithetical to . . . kinder, more humanistic approaches to work. Management approaches like Musk's threaten current business management practices that advocate for healthy, happy and engaged workplaces. Musk adheres to a mechanistic style of management that treats employees like cogs in a machine, rather than human beings" [45] (paras. 5–6).

While the improper handling of mass layoffs, setting of unreasonable work expectations, and pitting of employees against one another subsume a series of toxic leadership behaviors under a general management attitude of impertinence, Elon Musk is not above targeting individual employees with the same degree of insensitiveness and cruelty. Johana Bhuiyan reported that, "Musk publicly announced the termination of an engineer named Eric Frohnhoefer, tweeting 'he's fired' in response to Frohnhoefer's tweet correcting an assessment Musk made about why the site was so slow" [42] (para. 9). Elon Musk also mocked a worker with a disability (Haraldur Thorleifsson), tweeting on 7 March 2023, "The reality is that this guy (who is independently wealthy) did no actual work, claimed as his excuse that he had a disability that prevented him from typing, yet was simultaneously tweeting up a storm. Can't say I have a lot of respect for that". Likely trying to avoid a defamation lawsuit, Musk later deleted that tweet.

These examples suggest that Elon Musk, who is "notoriously thin-skinned" [46], despite claiming to be a free-speech absolutist is, like Donald Trump, intolerant of any kind of criticism. Specifically, Elon Musk sued the Center for Countering Digital Hate (CCDH), a nonprofit anti-hate research group that found hate speech had proliferated on the platform since he took over [47], banned journalists from Twitter who were critical of him [48], fired several employees who tweeted corrections to or countered things he has said on Twitter [42], and "in one case publicly called out a former employee's tweets about him saying that they were the result of 'a tragic case of adult onset Tourette's'" [49] (para. 8). As Ellen Pao wrote in The Washington Post, "Musk . . . often punches down in his tweets, displaying very little empathy. He called a British caver who helped to rescue trapped young Thai divers 'a pedo guy' (beating a defamation suit over the slur but adding to his reputation as a bully)" [50] (para. 3).

While Elon Musk appears willing to bully, intimidate, and potentially fire anyone who is critical of him, he has demonstrated a particular insensitivity on matters of gender and diversity [51]. On 21 June 2023, Musk tweeted, "The words 'cis' or 'cisgender' are considered slurs on this platform." Two months earlier, Twitter had removed protections for transgender people from its hateful conduct policy. As Clare Duffy reported at CNN:

> Twitter appears to have quietly rolled back a portion of its hateful conduct policy that included specific protections for transgender people. . . . Twitter also removed a line from the policy detailing certain groups of people often subject to disproportionate abuse online, including "women, people of color, lesbian, gay, bisexual, transgender, queer, intersex, asexual individuals, and marginalized and historically underrepresented communities". [52] (paras. 1–2)

We close this section with this point because there appears a notable confluence between toxic leadership, the rise of authoritarianism (at least, in Western contexts), and white heterosexual patriarchy. In fact, perceived threats to the hegemony of "whiteness" and "masculinity" in some spheres is likely a key contributing factor to the evolution of toxic leadership in the direction of digital authoritarianism. That perception was certainly central, as Brian Ott and Greg Dickinson noted, to Donald Trump's presidential victory in 2016. In their view, "The commonality between communication practices and communication platform [struck] a powerful emotive chord with [Trump's] followers, who [felt] aggrieved at the decentering of white masculinity" [1]. Basically, authoritarians' followers are drawn to speech and platforms where they can say anything they like without consequence (at least for them).

### 4.4. Impulsivity: Erratic and Unpredictable Decision Making

The fourth and final digital logic is impulsivity. It is a habit of mind that favors the rash over the reasoned and the affective over the rational. The logic of impulsivity leads to erratic, irregular, unpredictable, and unreliable decision making. Paradoxically, this habit of mind arises from humans' desire—but inability—to replicate the guiding principle of digital media, namely ever-increasing computational power and efficiency. Digital communication technologies, which are powered by microprocessors, are engineered to be as efficient as possible. Simply put, they are designed to perform as many mathematical calculations as rapidly as they can. Each generation of microprocessor possesses greater computational power, meaning it is more efficient.

While computers are highly efficient at receiving information, processing it, and executing commands, humans are comparatively less so. "That difference matters", explains Ott:

> because repeated exposure to the logic of efficiency formally invites humans to strive for greater efficiency. As humans attempt to mimic computer efficiency, they rely more heavily on instinct and affect. In short, as humans try to speed up their information processing and decision-making capabilities, they are less careful and rational and more impulsive and affective, which . . . undermines the quality of their decision making. [23] (p. 10)

In management and leadership contexts, this habit of mind routinely results in impulsive and ill-considered decisions. Moreover, it regularly enlists the destructive behaviors of misleading, lying, and a lack of transparency along with the dysfunctional personality traits of arrogance, deceitfulness, and untrustworthiness to justify those decisions. After all, digital authoritarians do not see themselves as responsible for their decisions thanks to the logic of impertinence. Further, the decisions arising from the logic of impulsivity are not only rash and reckless, but they are also simple-minded and reductionistic thanks to the logic of intransigence. Collectively, the digital logics of intransigence, impertinence, and impulsivity lead to poor decision making, which leaders, paradoxically, regard as effective thanks to the positive user feedback that often accompanies the logic of publicity.

The logic of impulsivity is a hallmark of Elon Musk's management of Twitter. As Walter Isaacson shared with The Wall Street Journal in an excerpt from his new book on the mercurial leader, "The way Musk blustered into buying Twitter and renaming it X was a harbinger of the way he now runs it: impulsively and irreverently" [53]. This habit of mind is evident in everything from his decision to purchase Twitter and subsequent attempt to back out of the deal [27]; to his ending of the legacy blue checkmark system and multiple restarts of a new paid checkmark system [32]; to his banning of linking to external social media sites and reversal of that decision [54]; to his limiting of how many tweets users can

view and changing of that limit multiple times in a few hours [55]. The near instantaneous reversal and/or revision to these decisions highlights their impulsivity, leading journalists to routinely describe the situation at Twitter as "chaotic" [43], "chaos" [44,56], "chaos and confusion" [57], and "widespread chaos and turmoil" [45].

Since Elon Musk's leadership of Twitter is a product not simply of his decisions, but also how the media covers his decisions, we share an extended quote from Ryan Mac et al.'s assessment of Elon Musk's purchase of Twitter in The New York Times:

> To a degree unseen in any other mogul, the world's richest man acts on impulse and the belief that he is absolutely right. ... As Twitter negotiated a sale to Elon Musk last month, the social media company pulled out a corporate takeover playbook. Mr. Musk, the world's richest man, did the opposite. He had no plan for how to finance or manage Twitter, Mr. Musk told a close associate. . . . And when Twitter resisted his overtures, Mr. Musk pressured the company with a string of tweets—some mischievous, some barbed and all impulsive. [58] (para. 5)

The preceding assessment is intriguing because it highlights that not only is Elon Musk's management style and decision making impulsive, but so, too, are his tweets. This is consistent with research that suggests "impulsivity" is a defining structural logic of both digital media generally and the platform of Twitter specifically. As Brian Ott explains, "Tweeting . . . requires almost no effort . . . It is ridiculously easy. Thanks to wireless technology, one can tweet from virtually anywhere at any time. Since tweeting requires little effort, it requires little forethought, reflection, or consideration of consequences. Tweeting is, in short, a highly impulsive activity" [59] (p. 61).

Because tweeting is impulsive, it invites ill-advised tweets such as Elon Musk's (since deleted) tweet perpetuating a "fringe conspiracy theory about the violent attack on Paul Pelosi" [60] (para. 1). The danger of this tweet is not simply in its recirculation of misinformation, though it accumulated more than 28,000 retweets and 100,000 likes before Musk deleted it. It is also dangerous—from a leadership perspective—because Elon Musk has chosen to manage Twitter (at least part of the time) on/through Twitter, making it impossible to meaningfully separate Twitter CEO Musk from private citizen Musk. Once a leader engages in management behaviors on social media, all their behavior on social media becomes part of their public record of management.

The implosion of management and nonmanagement spheres can, in light of the intersecting logics of publicity and impulsivity, become especially messy and fraught, as it did on 7 November 2022, when Elon Musk tweeted: "My commitment to free speech extends even to not banning the account following my plane, even though that is a direct personal safety risk". Musk's tweet referred to @ElonJet, a popular Twitter account that tracked and reported the movements of his private Gulfstream G700 jet in real time. Since the account regularly tracked short, 40-mile flights from San Franscisco to San Jose, it undermined Musk's "environmentally-friendly image" as CEO of an electric car company [26], which our previous analysis would suggest likely did not sit well with him.

About four weeks following Musk's tweet making "a big deal about how he wouldn't ban @ElonJet because of his 'commitment to free speech'" [61] (para. 4), Jack Sweeney, the account's owner, shared that an anonymous Twitter employee told him that Twitter had been shadowbanning @ElonJet, which severely limited its visibility. After Sweeney made this news public, Twitter "reversed course and reinstated @ElonJet's visibility" only to reverse the decision again and take the far more drastic step several days later of suspending the account altogether.

This series of events captures well, we maintain, the dangers associated with the intersecting logics of digital authoritarianism. Elon Musk publicly and perhaps impulsively tweets something that he believes makes him look good: his supposed staunch commitment to free speech in a context involving him personally. But privately, he engages in management behavior that punishes a private citizen and non-employee for causing personal offense. When a whistleblower at Twitter exposes this behavior, Twitter temporarily

reverses the shadowban to avoid the appearance of Musk targeting a private citizen. Then, after a few days have passed and people have hopefully moved on, Twitter suspends the account permanently.

Authoritarians do not recognize any limits or bounds to their authority. Because they exercise their authority publicly, they regard the entire public as subject to their authority. This is what it means to say, as we argue in the implications, that the harms of digital authoritarianism are social as well individual (for employees) and institutional (for the organization).

## 5. Critical Implications

As our analysis of Elon Musk's management of Twitter illustrates, the destructive behaviors and dysfunctional personality traits of toxic leadership coalesce around the four digital logics of publicity, intransigence, impertinence, and impulsivity, thereby transforming toxic leadership from a mode of abusive incompetence into outright authoritarianism. In the final section of our essay, we reflect on the implications of that evolution, highlighting (1) what it means for the perceived stability of traditional categories of leadership, as well as their treatment in academic literatures; (2) the value and importance of attending to the changing nature of communication technologies in leadership research; and (3) the individual, institutional, and social harms that accompany the spread of digital authoritarianism.

### 5.1. On the Evolution of Toxic Leadership

As our analysis demonstrates, styles of leadership are not static. Rather, they evolve over time to address our changing social environment. In this essay, we examined how toxic leadership is being remade by the transition from an analog age to a digital one. In an analog era, the impact of toxic leadership was limited largely to the organizations and employees who had toxic bosses. In that context, it made sense to theorize toxic leadership chiefly in terms of followership. But in a digital world, which obeys the controlling logic of publicity, toxic leaders increasingly enlist "publics" in their cause.

As such, it no longer makes sense to approach toxic leadership, or any style of leadership for that matter, primarily through the lens of followership, as least not as that category has traditionally been understood since "followers" no longer refers exclusively or even principally to employees. Moreover, at the same time that the logic of publicity has collapsed the personal, professional, and public, the related digital logics of intransigence, impertinence, and impulsivity have gradually cultivated authoritarian tendencies in those who occupy positions of leadership. As P. D. Harms and Karen Landay have concluded, "Whether conceptualized as a set of values, attitudes, or a personality trait, authoritarianism is associated with a variety of characteristics, including dogmatic, inflexible thinking [,] . . . intolerance of and lack of empathy for [others, and] . . . lower cognitive abilities" [62] (p. 164). Apart from impulsivity, it is striking how closely these authoritarian characteristics map onto digital logics.

Additionally, just as it no longer makes sense to approach toxic leadership through the frame of traditional followership, we question the wisdom and appropriateness of continuing to treat authoritarianism as a valid and legitimate category of leadership in academic literature. Authoritarianism is obviously an important social phenomenon, and therefore worthy of academic study. But given that it does not involve anything resembling recognizable and agreed upon leadership practices, treating it as a valid style of leadership legitimizes its abusive management practices in business and nonbusiness contexts. At best, the concept of authoritarian leadership is oxymoronic, and, at worst, it is unethical.

We want to acknowledge that there is a body of literature that argues "authoritarian leadership" is effective in some contexts. For example, scholars have documented its positive effect on employee motivation in some contexts and have argued the case of an "authoritarian advantage" in crises response [63,64]. Our concern is not with the conclusions of this scholarship, but with the terminology being employed. There are of course some

contexts where authoritarian practices and behaviors are sensible and warranted. But referring to these contexts in terms of leadership suggests a level of follower choice and power-sharing that simply does not exist. Scholars across many disciplines recognize that words matter, and the words we use in our scholarship matter as well.

In that spirit, we would slightly revise our own claim in this essay. Digital authoritarianism is not so much a new style of leadership as it is the transformation of toxic leadership into a form of rulership by persons in positions of authority. If we truly wish to take a stand against the growing authoritarianism in our world not just in political contexts but also in business, educational, and religious contexts, then we can start by passionately insisting that there is no context in which authoritarianism constitutes leadership.

### 5.2. On the Importance of Media Ecology

In her research on toxic leadership, Jean Lipman-Blumen stresses the significance of social change and the uncertainty and unease it creates as important factors contributing to the followership of toxic leaders [8]. However, she stops short of identifying the drivers of social change and the implications of those drivers, stating simply, "As humans, we are constantly bombarded by uncertainty, change, turbulence and crises" [8] (p. 3). Social change and its consequences, however, are not unmappable. In this essay, consistent with media ecology, we suggested that prevailing media forms or technologies of communication are the primary drivers of social change because they create the environment that we inhabit.

As Lance Strate explains, "our . . . modes of communication, our technologies and techniques, and inventions and innovations . . . have had a profound impact on human affairs, arguably greater than any other factor [,] . . . media . . . constitute the conditions we live under, the conditions we live in [,] . . . conditions that in turn condition us" [65] (para. 5). Another media ecologist, Jack Goody, made a similar point decades earlier: "Systems of communication are clearly related to what [humans] can make of [their] worlds both internally in terms of thought and externally in terms of . . . social and cultural organization. So changes in the means of communication are linked in direct as well as indirect ways to changes in the patterns of human interaction" [66] (p. 3).

Consequently, scholarship on leadership needs to take seriously the insights of media ecology. While the focus of the present essay was on identifying and analyzing the influence of digital logics on toxic leaders and leadership, future research could employ this perspective to better understand the shifting terrain of followership noted earlier. After all, the digital moment and its attendant logics have created a dangerous and deleterious situation in which social media users, who often know nothing about how an organizational decision was made or executed, publicly celebrate and endorse it. In the case of digital authoritarianism, this problem is greatly magnified by the cult of personality surrounding figures like Elon Musk and Donald Trump, who can do no wrong in the eyes of their followers.

In 2018, The Verge published an exposé on "Muskateers", which Bijan Stephen described as, "a vast, global community of people who revere the 46-year-old entrepreneur with a passion better suited to a megachurch pastor than a tech mogul" [67] (para. 5). Muskateers are not "followers" in the academic sense of the term, or, at least, not as used in leadership and management contexts. They are, for the most part, not employees. Muskateers are followers in the literal sense of the term in that they "follow" Elon Musk on various social media platforms; but they are also followers in an ideological sense, meaning they are loyal to Musk in much the same way Trump's supporters are loyal to him: unconditionally. As such, they are always "at the ready" not only to defend him, but also to attack those who are critical of him. Wrote Bijan Stephen:

> The most vocal of those fans have an impact: they're an army of irregulars waiting to be marshaled via a tweet and sent on the digital warpath against anything Musk decides he doesn't like, the iron fist in Musk's velvet glove. They've become known for haranguing people they believe have crossed him, journalists especially, with relentless fervor. The attacks are standard social media-era fare: free-for-all bombardment across social platforms by people who are not

always vitriolic but who nevertheless barrage the perceived enemy with bad-faith questions. As is often the case on these platforms, if you're not a cis white man, the harassment scales proportionally based on how far you deviate from that perceived norm. [67] (paras. 6–7)

The apparent intersection here between authoritarianism (with its cult of personality) and white masculinity deserves closer scrutiny. As critical management scholars investigate that intersection, they would do well to be mindful of prevailing communication technologies and their attendant habits of mind.

### 5.3. On the Individual, Institutional, and Social Harms of Digital Authoritarianism

Digital authoritarianism reflects a uniquely dangerous evolution in toxic leadership, subsuming the destructive behaviors and dysfunctional personality traits of toxic leadership to the structural logics of digital media (i.e., publicity, intransigence, impertinence, and impulsivity). This constellation is dangerous because it moves toxic leadership in the direction of authoritarianism, which inflicts serious and enduring harms not only upon persons and organizations as toxic leadership does [10], but also upon society in general. Given the breadth and depth of these harms, it is worth reflecting upon them at length.

At an individual level, digital authoritarians heighten the "conflicts and emotional damage to their subordinates" created by toxic leaders [7] (p. 66). Such damage takes a debilitating toll, not only on the careers of said subordinates, but also on their physical and mental wellbeing. Scholars have documented such psychological distress in the form of agitation, withdrawal, and loss of self-worth [7], as well as anxiety, fear, and depression [68]. As Birol Başkan indicated, "individuals within the organization [begin] to be treated as objects needed for the goal [...] instead of assets that facilitate the achievement of the organizational goal" [12] (p. 101). One need not have more than a basic understanding of human needs to understand the harms of being treated as an object.

At an institutional level, digital authoritarians negatively impact a number of key organizational outcomes. For example, authoritarian leadership negatively affects overall organizational performance [69], employee creativity [70], and turnover [71]. Perhaps more relevant to our arguments here, authoritarian leadership may also lead to counterproductive behaviors among subordinates. As Sunita Mehta and G. C. Maheshwari explained, "counterproductive behaviors tend to be attributed to perceived injustice by employees who retaliate by inflicting harm and producing systemic damage in an organization like sabotaging operations, providing inaccurate information, and being uncooperative to co-workers" [72] (p. 21).

At a societal level, digital authoritarians normalize incivility, undermine democratic norms and institutions, and even provoke violence. We want to concentrate on the last harm, which suggests potential links between leadership and violence. As the science fiction writer Isaac Asimov once observed, "Violence is the last refuge of the incompetent" [73]. Given that digital authoritarians are incompetent and abusive leaders who are interested only in power, they will do nearly anything to remain in power. Donald Trump's fomenting of an insurrection at the US Capitol following his defeat in the 2020 US election is a prime example. Importantly, while Trump did not engage in violence himself, he created a context in which his followers regarded violence as an acceptable and reasonable response.

The link between a leader and violence need not be nearly so explicit and direct, however. Some followers, especially those in the public sphere who were not subject to the abusive behaviors of a toxic leader (e.g., they are not employees), may take it upon themselves to respond to critics of a leader in a variety of abusive and violent ways including online harassment, mobbing, and doxing. But there is yet another potential link between digital authoritarianism and violence, which is attitudinal mimicry.

We worry that employees who engage in workplace violence may, in some instances, be taking their cues from years of toxic and abusive leadership in their organizations. If the leader of an organization consistently models impertinent behavior that suggests the ideas and opinions of others, but especially subordinates, do not matter, will employees begin to

internalize attitudes that similarly show little regard for others? Moreover, if "attitudes" are, as Kenneth Burke, argues, "incipient acts" or "the first step towards an act" [74] (p. 236), does an impertinent attitude increase the likelihood of workplace violence? More research on these potential links is desperately needed.

*5.4. Closing Thoughts*

As the preceding discussion stresses, the implications of digital authoritarianism could scarcely be more serious. Media ecologists have long recognized that our prevailing communication technologies shape and condition our habits of mind. That insight has historically been used to understand the broad social differences between various eras such as orality and literacy. But in this essay, we suggested that organizational and management scholars can also benefit from the insights of media ecologists by studying how our digital communication environment privileges the logics of publicity, intransigence, impertinence, and impulsivity.

Specifically, we argued that these logics transform toxic leadership into a dangerous form of authoritarianism that places the narcissistic needs of a leader ahead of their organization, favors simple and inflexible decision making, demonstrates little regard for the wellbeing of employees, and produces erratic and ill-considered decisions. Only by understanding the structural biases of media, and then making critically conscious decisions regarding habits of mind, can we begin to envision ethical models of leadership in our digital world.

**Supplementary Materials:** The following supporting information can be downloaded at: https://www.mdpi.com/article/10.3390/world4040046/s1, Elon Musk Leadership Action Chart.

**Author Contributions:** Conceptualization, B.L.O. and C.S.H.; methodology, B.L.O. and C.S.H.; formal analysis, B.L.O. and C.S.H.; investigation, B.L.O. and C.S.H.; data curation, B.L.O. and C.S.H.; writing—original draft preparation, B.L.O.; writing—review and editing, B.L.O. and C.S.H. All authors have read and agreed to the published version of the manuscript.

**Funding:** This research received no external funding.

**Institutional Review Board Statement:** Not applicable.

**Informed Consent Statement:** Not applicable.

**Data Availability Statement:** Not applicable.

**Conflicts of Interest:** The authors declare no conflict of interest.

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
