# Peer review of "The Digital Authoritarian: On the Evolution and Spread of Toxic Leadership"

_world, doi:10.3390/world4040046_

Round 1

Reviewer 1 Report

Comments and Suggestions for Authors

The topic is interesting, but the paper needs some amendments.

First, authors need to develop and add clearly states research questions of the paper. Subsequently, this research questions should be clearly addressed in the concluding chapter, which is actually completely missing.

Second, I would recommend that authors restructure the paper to follow IMRaD structure, as this is applicable given the nature of the study. 

The recommendation is that authors uniform the utilization of names and surnames, because sometimes they state both, sometimes just surname, sometimes Mr. and then surname. This should be standardized.

Author Response

Please see attached reply.

Reviewer 2 Report

Comments and Suggestions for Authors

The paper guides the reader through an exposition on toxic leadership in the digital frame, with Elon Musk as subject. The paper is quite entertaining and makes a number of excellent points of what should be frightening consequence for readers. I don't know that the material is 'new' as much as eloquently brought together and expertly argued here. I do have some points of concern/clarification; I don't consider them major but worthy of response -

First, the paper does appear to begin with an end in mind - it is not all that balanced but I don't know that it has to be to succeed as a paper, but it does have to own it's lack of balance. Some comment about it is made but perhaps not enough? Not everyone sees Elon Musk as a problem or toxic or an authoritarian. Some see him as a maverick or some kind of hero (I don't, but still, others do, and they buy his cars and look with awe at his space ventures).  I think the points about his shambolic run of Twitter/X do well enough to demonstrate his lack of competence in management, but many give him a pass as an exceptional leader and still do.  Like Trump, he is a polarizing figure - Trump is either a shrill, insulting carnival barker/clown or America's greatest President. What a time to be alive. Musk is not merely misunderstood; he is not 'deeply flawed' but an exceptional human being - and authoritarianism of the type you mention likely does deserve discussion in an academic piece.  In a way this reminds me a bit of Adorno on authoritarianism, mentioned more recently in Jones, B. (2022). America’s Authoritarian Slide. Political Insight, 13(1), 34–36. https://doi.org/10.1177/20419058221091637

To this point, you seem to indicate that democracy is declining, but is democracy the aberration, and authoritarianism more the norm?  Over the very long term? Is Musk just another symptom of the sickness?

But I wonder about the lack of 'method' in this inquiry - it's an essay, it is well argued, but the authors might have explained better their approach and why they chose to examine what they've examined here. I think it's great that some points are brought up, especially the hostility to research. But there must have been some plan for including and not including material. It's not necessarily clear to me what that was.

The issue of perception of Musk's leadership is raised, and frankly, I don't even see that as an issue.  He's a public figure, he posts constantly and wants everyone to know what he thinks, then he cannot own the perception the public has of him, even if he desperately is trying to control them anyway.

Have you given thought to the idea that the whole twitter rebrand might have been stolen in the first place? Does Musk have original ideas, or does he just borrow?  https://www.cnbc.com/2023/07/25/elon-musk-twitter-rebrand-x-legal-trouble-with-meta-microsoft.html

I like the digital authoritarianism angle on this - it's a useful (if scary) contribution and the paper is worthy of further consideration, but some response to the above would be helpful.

Author Response

Please see attached response.

Reviewer 3 Report

Comments and Suggestions for Authors

Dear Authors,

Thank you for the opportunity to review your manuscripts. Although there are a lot of interesting things in it, there are some comments that I think you will take into account.

1. It is not clear from the abstract what the genre of the manuscript is. Said to be an essay, but similar in structure and scope to an article.

2. I am not sure that the line manager at Starbucks really does as stated in lines 36-38. Wanted evidence and links to the source.

3. A very bold and somewhat superficial description of digital authoritarianism. Why this management style becomes toxic leadership. Maybe I would suggest taking inspiration from classical descriptions. For example, inspiration can be found here: Steblyna, N., & Dvorak, J. (2021). Reflections on the independent mass media of post-Soviet countries and political competitiveness. Politics in Central Europe, 17(3), 565-588.

4. The introduction lacks the goal, and the research question, since the text does not contain a description of the methodology, it is not at all clear why the case of Elon Musk's leadership was chosen, how it was selected, and what other cases were considered.

5. It is also unclear what the connection is between Elon Musk's management and public management. Want more arguments?

6. It is not very clear why the characteristics described in lines 89-95 are unique to a toxic leader, it is also characteristic of a corrupt person, what does this paragraph have in common with a toxic leader?

7. It is not very clear why in Table 1 the elements for the example of intransigence are given under digital technology; impertinence; impulsivity; and publicity is good for being there because it seems that they are good for toxic leadership. And an element of digital technology can be a social media property.

8. I wish that the critical analysis of section 4 was connected with the methodology and why such frames were chosen for the analysis.

9. Section 5 would be more correct to transform into a discussion, conclusions and implications. 

All the best

Author Response

Please see attached response.

Round 2

Reviewer 3 Report

Comments and Suggestions for Authors

Dear Authors,

Many thanks for the open and clear explanation of your approach and way of thinking. I think your manuscript can be accepted to the World journal.

All the best